# Virtual Screening and Validation of Affinity DNA Functional Ligands for IgG Fc Segment

**DOI:** 10.3390/ijms25168681

**Published:** 2024-08-09

**Authors:** Qianyu Yang, Zhiwei Liu, Xinrui Xu, Jiang Wang, Bin Du, Pengjie Zhang, Bing Liu, Xihui Mu, Zhaoyang Tong

**Affiliations:** State Key Laboratory of NBC Protection for Civilian, Beijing 102205, China; qyyang918@163.com (Q.Y.); xuxinrui710@163.com (X.X.); roverman@163.com (J.W.); dubin51979@163.com (B.D.); zpjbit@163.com (P.Z.); lbfhyjy@163.com (B.L.); muxh0511@163.com (X.M.)

**Keywords:** DNA functional ligand, molecular docking, virtual screening, Fc segment, affinity

## Abstract

The effective attachment of antibodies to the immune sensing interface is a crucial factor that determines the detection performance of immunosensors. Therefore, this study aims to investigate a novel antibody immobilization material with low molecular weight, high stability, and excellent directional immobilization effect. In this study, we employed molecular docking technology based on the ZDOCK algorithm to virtually screen DNA functional ligands (DNAFL) for the Fc segment of antibodies. Through a comprehensive analysis of the key binding sites and contact propensities at the interface between DNAFL and IgG antibody, we have gained valuable insights into the affinity relationship, as well as the principles governing amino acid and nucleotide interactions at this interface. Furthermore, molecular affinity experiments and competitive binding experiments were conducted to validate both the binding ability of DNAFL to IgG antibody and its actual binding site. Through affinity experiments using multi-base sequences, we identified bases that significantly influence antibody-DNAFL binding and successfully obtained DNAFL with an enhanced affinity towards the IgG Fc segment. These findings provide a theoretical foundation for the targeted design of higher-affinity DNAFLs while also presenting a new technical approach for immunosensor preparation with potential applications in biodetection.

## 1. Introduction

Biosensors employ immobilized biological sensitive materials on the sensing interface as recognition elements, converting biochemical molecular information into acoustic, optical, and electrical signals through a transducer for real-time detection in biomedicine, environmental monitoring, food industry, and disease diagnosis [1,2,3,4]. Enhancing the density of immobilized antibody molecules on a unit area transducer (sensor chip) surface while preserving their activity at the biosensing interface is crucial for improving the sensitivity and selectivity performance of biosensors [5,6,7]. Traditional methods such as physical adsorption and covalent bonding interactions may lead to reduced antibody activity and a lack of control over antibody orientation on the sensing interface. In some cases, fragments of antigen binding (Fab) that specifically recognize antigens may randomly bind to the sensing interface, resulting in antibody inactivation [8,9]. Self-assembled membrane (SAM) technology offers high orderliness, adjustability, and thermodynamic stability which partially addresses issues related to reduced antibody activity caused by traditional methods. When applying the self-assembling membrane method to the immunosensor chip, it is necessary to modify the antibody with an amino or carboxyl tail in order to bind the functionalized self-assembling group. However, determining the position of active group modification on the antibody is generally challenging, thereby limiting the ability of the SAM method to ensure proper orientation of antibodies on the film and consequently restricting exposure of Fab recognition site outside the sensor chip, which impedes improvements in sensor sensitivity and specificity [7,10]. Therefore, it is imperative to develop antibody immobilization materials that specifically bind to non-antigen recognition fragments, such as fragment crystallizable (Fc), and enable the directional and ordered arrangement of antibodies at the sensing interface to address the challenge of efficient antibody-antigen recognition in immunosensors.

Previous studies have demonstrated that staphylococcal protein A (SPA) can specifically bind to the Fc fragment of immunoglobulin G (IgG) antibodies through intermolecular forces. Simultaneously, the Fab fragment remains exposed externally to achieve directional immobilization of the antibody at the immune sensing interface without compromising its activity [11,12,13]. Since its discovery, SPA has found extensive applications in antibody purification, affinity chromatography, immunoassays, biosensors, and other fields [14,15,16,17]. The excessive steric hindrance of SPA will impact the immobilized density of antibodies, thereby influencing the sensitivity of the sensor, and the high costs associated with production and preparation still pose a challenge in implementing SPA. Consequently, SPA is not an ideal or optimal material for immobilizing antibodies in immune detection.

The deoxyribonucleotide molecules, known as DNA, play a pivotal role in the transmission of genetic information and serve as essential components in various biological processes. Owing to its distinctive functional groups, DNA can be the ligands to form complexes with numerous macromolecular proteins through covalent or non-covalent interactions. The single-stranded deoxyribonucleotide (ssDNA) fragment possessing specific binding capabilities within these complexes is referred to as DNAFL [18]. Traditional DNAFL (such as gene probes) can make up nucleotide double strands with their complementary target DNA or RNA sequence by nucleic acid hybridization, and the specific target genes can be detected and analyzed for expression of bacteria, viruses, and other microorganisms [19,20]. Another significant application of DNAFL lies in nucleic acid aptamers that are primarily obtained through systematic evolution of ligands by exponential enrichment (SELEX) [21] who differ from previous examples in that they utilize base sequences to form distinct three-dimensional spatial structures for target recognition purposes, and their targets include proteins, peptides, bacteria, viruses, small organic molecules, and metal ions [22,23,24]. Due to its small molecular weight, minimal steric hindrance, straightforward preparation and modification process, cost-effectiveness, and robust stability, DNAFL can serve as a substitute for the conventional antibody directional immobilization material SPA [23,25]. This substitution enables the functionalization of immunosensors and facilitates their widespread application while potentially fostering the advancement of innovative biosensors.

Our research group has previously identified specific DNA sequences capable of binding to the Fc segment of antibodies, however, the precise mechanism underlying this binding interaction remains unclear. Consequently, virtual screening methods based on molecular docking have emerged for investigating the protein-ligand binding mode [26]. Molecular docking computation can simulate the orientation of protein and ligand binding sites, reveal intermolecular interactions, and evaluate affinity binding energy. Its applications primarily encompass the discovery and design of adjuvant drugs, identification of potential active ligands, and screening of bioactive targets [27,28,29]. Most of the current molecular docking procedures are based on multi-atomic systems of molecular mechanics, and the physical parameters involved mainly include potential energy, torsion term, bond geometry, classical term, and Lenard-Jones potential [30]. The process of molecular docking mainly consists of two essential components: sampling the protein binding sites near the active sites to generate acceptable ligand binding orientations and conformations, followed by evaluating the combination and ranking of ligand poses using score functions based on physical principles or empirical knowledge, in order to assess the affinity between the receptor and ligands [31,32]. However, the molecular docking method also encounters several challenges, for instance, the accuracy and precision of the docking method are significantly influenced by entropy and desolvation effects, and there is a need for further refinement for score functions [33]. Additionally, establishing a stronger correlation between docking results and experimental data remains a major challenge [34]. Protein-ligand molecular docking modeling methods have been developed based on molecular force fields which encompass three primary modes: rigid docking, semi-flexible docking, and flexible docking [35]. Chen et al. [36] developed the ZDOCK program (https://zdock.wenglab.org/, accessed on 24 July 2024) in 2002, which has become one of the main tools for studying proteins-ligand through rigid docking. This program employs a fast Fourier transform algorithm and enables efficient ligand sampling on a 3D grid. By considering factors such as shape complementarity, electrostatics, and other parameters, it achieves higher precision with comprehensive scoring [37]. However, the complexes generated by ZDOCK are not particularly suitable for designing protein interface inhibitors [38].

Biological layer interference (BLI), like surface plasmon resonance (SPR), has been developed for the investigation of intermolecular interactions [39,40]. The BLI involves coupling ligands to an optical fiber biosensor’s compatibility layer to increase the thickness of the biological layer. This leads to a detectable displacement curve in the reflected light interference spectrum, enabling real-time, free-label, and high-efficiency measurement of intermolecular interactions [41,42]. BLI is particularly suitable for kinetic analysis, detection, and quantification of DNA/RNA-protein interactions [43,44]. In biochemistry, the target and ligand interact to form a complex, with this reaction being reversible and gradually reaching a steady state in the buffer system. The strength of the interaction between the coupled ligand and target material is typically characterized by the equilibrium dissociation constant (*K_D_*), which quantifies the extent of complex dissociation at equilibrium. A higher *K_D_* value indicates increased dissociation, indicating a weaker affinity between the ligand and target, while a lower *K_D_* value suggests a stronger binding affinity [45]. The *K_D_* is not influenced by ligand and target concentration under the same conditions, such as temperature, pH, and salt concentration.

At present, there still exist challenges in achieving targeted immobilization of antibodies at the immune sensing interface. To replace the conventional SPA, which is characterized by high cost and molecular weight, it is crucial to develop small molecular materials that possess low molecular weight, cost-effectiveness, robust stability, and strong affinity for binding to the Fc segment of antibodies. In this study, we retrieved 140 DNAFLs structure and 1 whole molecular mouse IgG antibody from the Protein Data Bank (PDB) as ligands and receptors, respectively. The ZDOCK algorithm was utilized to perform molecular docking between the antibody and DNAFL, and the ligand targeting the Fc segment was screened based on its score and virtual binding mode. The interaction relationship between amino acids and nucleotides at the binding interface was determined by analyzing the critical binding sites and contact propensity. In addition, we conducted experiments to determine the binding affinity of DNAFLs with varying lengths and compositions to mouse IgG antibodies, as well as to validate their specific binding sites. Furthermore, we investigated the impact of different bases on this affinity. These findings will greatly facilitate the targeted design of functional ligands that exhibit higher affinities towards the Fc segment of antibodies, thereby providing valuable insights for developing innovative immunosensor interfaces for biodetection and designing high-performance biosensors with broad application potential.

## 2. Results

### 2.1. Molecular Docking

#### 2.1.1. Binding Mode in Molecular Docking Results

A complete antibody molecule consists of two fragments: the specific recognition antigen region (Fab) and the crystallizing region (Fc). 3D structure of an intact IgG2a monoclonal antibody with a PDB code of 1IGT [46] is depicted in Figure 1A, which primarily consists of two heavy chains (B, D), two light chains (A, C), and two oligosaccharide chains (E, F). The results of molecular docking revealed that, apart from the binding of DNAFL to the Fab segment of the antibody, the binding sites of other ligands primarily interacted with the Fc segment, which could be categorized into three distinct modes. The three modes of DNAFLs binding to an IgG antibody (with the binding domain shown in yellow) and the possible postures when immobilizing the antibody at the sensing interface by DNAFL are illustrated in Figure 1B.

DNAFLs binding situation in all three modes partially occupies a segment of the Fc structure, distinct from the fully bound to Fab, their binding sites are located within the non-antigen binding region. Mode 1 indicates complete binding of DNAFL to the Fc segment, while in Mode 2, the spatial structure of the hinge region can be easily substituted by other molecules, thereby facilitating limited amino acid binding for macromolecules such as DNAFL, meanwhile, the amino acids involved in Mode 2 are approximately located between LYS455 and LYS475. At last, the majority of DNA bases in Mode 3 are bound to the Fc segment, while the minority are bound to either the C_H1_ or C_L_ region (part of the Fab region).

#### 2.1.2. Binding Sites between IgG and DNAFL

After performing molecular docking of 140 single-stranded DNA (ssDNA) functional ligands with the IgG antibody, the highest ZDOCK score can be utilized to determine the binding of these ligands to the target region on the antibody. The sequence composition of these 140 DNAFLs, their respective binding modes with antibodies, and the corresponding ZDOCK score were presented in Appendix A, which was categorized based on different binding modes, clearly demonstrating that a total of 49 screened sequences highlights the specificity of the Fc segment. Notably, among them, the DNAFL derived from protein-ssDNA compound whose PDB code is 3HXQ exhibited an exceptional score of 32.78. To illustrate this point further, Figure 2 depicts the process and outcomes of molecular docking. The binding pose with the highest score out of 2000 possible poses is denoted as pose 1, as illustrated in Figure 2A, while a more intense red color signifies a higher score and a closer approximation to the most realistic binding conformation. DNAFL (highlighted in yellow) is situated within the C_H_2 region of the chain B and D, specifically within the Fc segment of the antibody, as depicted in Figure 2B, it occupies the uppermost part of the central space, and this region consistently emerges across all molecular docking outcomes involving DNAFL and antibodies with the binding process involves the participation of carbohydrates.

The binding frequencies of each amino acid located in the Fc segment of the B and D chains are depicted in Figure 3A, and the primary binding site for the former encompasses Phe254-Lys261, Arg353-Lys360, Pro397-Glu405 and Tyr419-Pro424, while for the latter it comprises Pro257-Ile260, Glu398, Glu399 and Glu423-Leu426. Both chains B and D possess identical amino acid composition and exhibit similar spatial structure, and they demonstrate nearly equivalent tendencies for nucleotide binding, however, the former exhibits a higher overall frequency. On the other hand, binding frequencies of individual monosaccharide sites are illustrated in Figure 3B, mainly involving three loci (Nag8, Man7, and Bma3). The left panel of Figure 3C illustrates the distribution of high-frequency amino acids, with a color gradient from yellow to red indicating an increase in frequency while the right panel highlights the specific position of Glu423 on both strands. The observation further substantiates the significance of the amino acids surrounding the cavity, as well as the two sugar chains.

The appropriateness of molecular docking was assessed by mutating Glu423, the highest binding amino acid site on the heavy chain B, to glycine and redocking it with DNAFLs from 3HXQ, 5XS0-Y and 4I28 whose scores are highest, intermediate, and lowest score, respectively. The distribution of binding sites before and after mutation is illustrated in Figure 4, revealing no significant alteration in the binding position upon re-docking. Notably, Glu423 residue remains engaged with these three DNAFLs even after being substituted with glycine. Furthermore, their ZDOCK scores decreased from 32.78, 20.34, and 17.18 to 29.1, 18.42, and 16.58, respectively, indicating that the glutamic acid site facilitated nucleotide chain binding.

#### 2.1.3. Contact Propensity of Protein Units to Bases

To comprehend the impact of a structural unit in the protein on nucleotide binding, we calculate the contact tendency *P_ij_* (defined by Equation (1)) between amino acids and monosaccharides on the binding interface of the virtual complex with each base in the nucleotide [47]. Here, *N_ij_* represents the total number of contacts between base *i* and amino acid residue (or monosaccharide) *j*, *T_j_* denotes the total number of contacts for residue (or monosaccharide) j in the entire dataset, and a cut-off distance ranging from 3–6 Å is set for different types of non-bonded interactions.
(1)$$P_{ij}=\frac{N_{ij}}{\sum N_{ij}}/\frac{T_{j}}{\sum T_{j}}$$

According to the calculation, if *P_ij_* is less than 1, it indicates that the amino acid or monosaccharide is not favorable for interacting with the base; otherwise, the structural unit tends to interact with the base [48]. The interaction function of the protein binding interface was analyzed using DS, resulting in the determination of the binding tendency between 20 amino acids and 6 monosaccharides in the Fc segment with four bases (adenine, cytosine, guanine, and thymidine), as illustrated in Figure 5, which also presents the probability of each structural motive appearing on the binding interface of the area to be analyzed. Although Figure 5A indicates a strong tendency for mutual contact between the seven components (Ala-DC, Cys-DC, Gly-DA, His-DT, Met-DA, Gln-DT, and Trp-DA), with the highest *P_ij_* value reaching even up to 1.73, the amino acids they belong to are rarely observed in the binding site of the Fc segment and these amino acids were not considered in subsequent studies. Similarly, the monosaccharides in the oligosaccharide chain also exhibit a pronounced propensity to interact with bases, and Figure 5B illustrates that N-acetylglucosamine (44.6%), which is most frequently found at the junction, displays a higher likelihood of contacting guanine. The galactose-cytosine pair exhibits the highest *P_ij_* value of 2.19, however, it only has a 6.6% probability of being present at the interface between the antibody and nucleotide chain.

### 2.2. Affinity and Sites Validation

#### 2.2.1. Affinity and Sites Validation of Virtual Screening DNAFLs

The Octet K2 system can be utilized for the determination of kinetic measurements, including the association rate constant (*K_on_*), dissociation rate constant (*K_dis_*), and *K_D_* of biomolecular interactions. These parameters are automatically generated by the Octet K2 Molecular Interactor. The default stoichiometric ratio of receptor to ligand binding is generally 1:1, with their interaction being determined by the kinetic constants *K_on_* and *K_dis_*, while *K_D_* can be calculated using Equation (2) [49], the units for *K_on_*, *K_dis_*, and *K_D_* are expressed in 1/Ms, 1/s, and M, respectively.
(2)$$K_{D}=\frac{K_{dis}}{K_{on}}$$

Upon investigating the affinity between mouse whole-molecule antibodies and DNA ligands, it was observed that 20 out of 49 DNA ligands exhibited significant affinity with mouse IgG molecule antibodies when bound to the Fc segment, and the corresponding *K_on_*, *K_dis,_* and *K_D_* are presented in Table 1. *R*^2^ represents the degree of fit, which generally needs to exceed 85%, and the higher the value of *R*^2^, the more reliable the affinity value becomes. To demonstrate the comparable affinity of DNAFL combining with the antibody as that of the SPA-fixed-antibody, *K_on_*, *K_dis_*_,_ and *K_D_* of SPA-IgG were also examined in this study as positive controls and are presented in Table 1. “Association-dissociation” curve is depicted in Figure 6A, where the ordinate represents the displacement distance of the interference spectrum curve, which directly correlates with the binding affinity intensity between them. When the displacement signal is ≥0.02 nm, it can be inferred that successful binding has occurred to the corresponding target. Among the 49 ligands tested, certain DNAFLs exhibited significant binding affinity while their notably low *R*^2^ rendered their interaction with IgG antibodies highly unstable. In the figure, DNAFL exhibited a remarkable binding signal of 0.206 nm at an antibody concentration of 0.2 mg mL^−1^, and the *K_D_* was determined to be 2.41 × 10^−7^ M using Equation (2). By directly comparing the *K_D_* values of the 20 sequences listed in Table 1, along with the DNA sequence information provided in Appendix A, it is possible to analyze the impact of variations in sequence length and base composition on affinity. The majority of ligands exhibited a *K_D_* value in the range of 10 × 10^−6^, while some ligands demonstrated an even lower affinity with a *K_D_* value as low as 10 × 10^−7^. It is obvious that bases G and T have a higher proportion in the sequence and that the length of at least 10 bases should be maintained for optimal affinity. In the Appendix A for this study, the “association-dissociation” curve of 20 affinity DNAFLs binding to mouse IgG is provided, moreover, it also includes the curves of SPA to IgG. It is noteworthy that the *K_D_* of SPA to mouse IgG was 1.49 × 10^−8^, which exhibited an approximately tenfold difference compared to the minimum *K_D_* (2.41 × 10^−7^) value of DNAFL to IgG, suggesting that while the selected DNAFLs can effectively bind antibodies, further refinement and screening are necessary in order to achieve higher affinity for DNAFL.

In this study, we utilized the affinity between DNAFL and antibody to investigate whether the DNA ligand can compete with SPA and bind to the antibody, thereby enabling detachment of the antibody from the surface of SPA. The results are illustrated in Figure 6B (protein compound 6GN7 as an example). The response signals depicted in the figure are all below 0.02 nm, indicating that the addition of DNAFL does not elicit any additional response signal after SPA binds the antibody to the biosensing interface through Fc binding, thereby suggesting that it fails to dissociate the antibody on SPA’s surface. The Appendix A also include “Association-dissociation” curves of 20 affinity DNAFLs competing with the SPA.

#### 2.2.2. Affinity of Poly-Bases DNAFLs

According to the data presented in Table 1 and Appendix A, it is evident that most sequence lengths yielding high-affinity fall within the range of 15 to 25, and the longer oligonucleotide chains may offer a greater number of amino acid affinity sites. Consequently, in order to comprehensively investigate the affinity between each base and the antibody, our focus was directed towards synthesizing nucleotide chains comprising a single base only, namely poly-bases and with varying lengths of 10, 15, 20, 25, and 30. Affinity assay (*K_on_*, *K_dis,_* and *K_D_*) of the poly-base series affinity with IgG antibodies can be found in Table 2, while Poly-A10 represents a DNAFL containing 10 adenine deoxyribonucleotides, and other sequences are also labeled in the same manner. Poly-A ligands exhibited no binding affinity towards the antibody protein irrespective of the length of the nucleotide chain, whereas the poly-C ligands demonstrated affinity only at lengths of 15 and 20 nt while remaining unresponsive at other instances. Nucleotide sequences of poly-G and poly-T, in contrast, exhibited lower *K_D_* indicating higher affinities towards IgG antibodies across all chain lengths. Notably, the highest affinities were observed at 15 nt and 20 nt with *K_D_* values of 2.73 × 10^−7^ and 8.30 × 10^−7^, respectively. However, their overall affinity gradually diminished as the sequence length increased and poly-G demonstrated slightly higher affinity compared to poly-T. After comprehensive consideration in this study, the binding-dissociation curves of poly-A, poly-C, poly-G, and poly-T are presented in Figure 7 for a sequence length of 20 nt, while poly-G20 is still an observed affinity of 3.12 × 10^−7^ under this situation.

## 3. Discussion

Antibodies are the most classical and widely used biological recognition molecules that directly determine the function and quality of immunosensors. Fc structure presented in 1IGT, the first murine Fc determined by X-ray crystallographic analysis, constitutes a complete anti-lymphoma antibody with 65% amino acid sequence identity to the human Fc fragment, devoid of insertions or deletions, and exhibits a significant level of amino acid sequence homology with the remaining portion. The Fc fragment located at the carboxyl terminus of this antibody is the site responsible for antibody-mediated immunity, exhibiting minimal variation and greater stability, which encompasses approximately half of the light chain C terminus and three-fourths to four-fifths of the heavy chain C terminus.

The study results revealed the absence of direct contact between the two oligosaccharide chains results in irregular cavities within the central region of the Fc segment. Additionally, DNAFL interacts with the surrounding amino acids to varying degrees through groove-shaped complementation achieved by rigid molecular docking. The optimal arrangement primarily involves immobilizing the antibody Fc segment onto the sensor chip surface in a systematic and oriented manner, ensuring that the Fab segment is positioned distantly from the sensing interface to fully expose the antigen recognition site. Among the three modes, only mode 1 enables optimal exposure of the antigen binding site and efficient directional immobilization of the antibody, while mode 2 and mode 3 may result in non-specific adsorption of antibodies onto the solid surface conversely which can reduce the effective attachment rate of interface recognition molecules and compromising sensor detection performance. Therefore, mode 1 represents the most ideal antibody fixation approach for immune sensing interfaces.

The virtual binding sites of DNAFL to IgG are predominantly localized around the central cavity of the Fc segment, exhibiting a distinct preference for the C_H_2 region of the chain B. Approximately 70% of DNAFL interacts with Glu423 on the B strand, which also exhibits the highest frequency on the D strand (28%), thereby indicating its significance as a nucleotide binding site within the Fc segment. Similarly, while there are slight differences in composition between the chains E and F, they both demonstrate a higher affinity for binding to chain B. The distribution pattern of amino acids and monosaccharide sites surrounding the cavity in the Fc segment resembles that observed in drug design pockets, suggesting that optimized DNAFLs can be designed based on this region’s shape and composition. Meanwhile, mutation of the site with the highest binding frequency did not result in an increase in the molecular docking score, which suggests that the binding sites identified using the original IgG antibody structure remain reliable, and further validate the appropriateness of the proposed docking protocol in this study.

Although proteins such as antibodies exhibit significant structural and functional diversity, there exist conserved preferences in the interactions between their residues and biological molecules like DNA. Consequently, numerous diverse approaches have been developed to investigate the interaction of amino acids within proteins with other molecules [50]. The binding sites exhibited a distinct preference for specific corresponding bases, with proline (13.1%), glutamic acid (10.2%), lysine (9.9%), and isoleucine (9.3%) being the most abundant amino acids involved in these interactions. Notably, Glu423, which displayed the highest binding frequency, demonstrated a stronger affinity towards guanine and there was minimal alteration in the binding propensity of these amino acids. Compared to the other three bases, proline, being the most frequently occurring amino acid, exhibits a higher propensity for interacting with cytosine deoxyribonucleotide. Conversely, glutamic acid, as the second most frequent amino acid, primarily interacts with guanine. Subsequently, lysine demonstrates a slightly stronger affinity towards purines compared to pyrimidines; however, the *P_ij_* values between them are both weak and similar. Isoleucine displays a comparable preference for adenine and cytosine but tends to establish more pronounced interactions with thymine. Overall, the amino acids involved in binding in the Fc segment exhibited a high affinity for guanine and cytosine, while their binding performance was relatively weak for thymidine and lowest for adenine. Among the top three monosaccharides (Nag, Man, and Bma) with the highest proportion in the oligosaccharide chain, guanine exhibits a strong affinity for binding (*P_ij_* = 1.13, 1.27 and 1.04, respectively), thus making it more likely for chains E and F to attract base G as a whole. Nucleotide sequences exhibiting a higher proportion of guanine enrichment are more likely to be found within the glycan region of the target-binding domain of the antibody when considered collectively. The abundance of binding sites and the interaction propensity between each amino acid and monosaccharide on the base will facilitate the directional design of specific DNA sequences for subsequent studies aimed at discovering DNA ligands with enhanced affinity.

The light interference signal emitted by the molecular interactor is converted into a real-time response signal on the surface of the SAX2.0 film interference biosensor based on the binding and dissociation behavior of molecules. Considering all factors collectively, the oligonucleotide strand derived from protein compound 6GN7 was identified as the ligand with the highest antibody affinity towards the antibody among the remaining 20 ligands exhibiting strong affinity. High-affinity DNAFLs shown in Table 1 are primarily composed of guanine and thymine deoxyribonucleotides, with a small amount of adenine and cytosine present, indicating their specific micromolar level of affinity. These findings suggest that nucleotides with higher guanine/thymidine ratios exhibit stronger binding to antibody proteins, which aligns with the results obtained from virtual docking experiments. It is worth noting that the sequences exhibiting affinity contain multiple poly-T sequences of varying lengths, and thymine is commonly found at the protein-DNA complex interface, thus indicating a certain level of affinity. The FC segment of the IgG antibody contains multiple high-affinity binding sites for SPA, which facilitates the specific attachment of the antibody while preserving its specific recognition ability. It can be inferred from Figure 6B that DNAFL and SPA possess similar binding sites when interacting with antibodies, particularly within their Fc segments. The results of this study provide further validation, demonstrating that none of the 20 DNAFLs listed in Table 1 exhibited any competing signals during the experiment which indicates a high likelihood for them to effectively bind to the Fc segment. To determine the precise binding sites of DNAFLs in the Fc segment of IgG antibodies, it is imperative to obtain the 3D structure of the complex formed by them through extensive crystal structure analysis, which requires further exploration. Finally, affinity tests of poly-bases DNAFLs can be further validated through the rich nucleotide sequence of guanine and thymine, which enhances their interaction with IgG antibodies in mice. DNAFL sequences within the range of 15 to 20 nt are considered optimal for achieving high-affinity binding, which conclusion establishes a reliable technical approach for designing high-affinity DNAFL and constructing targeted immune sensing interfaces.

## 4. Materials and Methods

### 4.1. Reagents and Apparatus

Single-stranded oligonucleotide chains (5′-end modified by biotin or without modification) were synthesized by Beijing XingFangyuan Biotechnology Co., Ltd. (Beijing, China); ChromPure mouse IgG whole molecule was purchased from Jackson Immuno Research (West Grove, PA, USA); Bovine serum albumin (BSA), glycine (0.1 M), and PBS (0.01 M) were purchased from Solarbio (Beijing, China); and Tween20 was purchased from Sigma-Aldrich (Shanghai, China). PBST-B (containing 0.02% Tween20 and 0.1% BSA) and regeneration solution (0.01 M glycine, pH = 1.70) were prepared in our lab. All the other reagents were analytically pure, and the water utilized was deionized.

Molecular docking and molecular dynamics simulation were performed by Discovery Studio Client 4.5.0.15071 (DS). The equilibrium dissociation constant (*K_D_*) was determined on Octec K2 Molecular Interactor 12.0.2.11 by ForteBio (Fremont, CA, USA). A thin film interference biosensor (TFI) was used. High-precision Streptavidin biosensors (sensor type: SAX2.0) and SPA biosensor (sensor type: ProA) were provided by Sartorius (Göttingen, Germany); deionized water was obtained using the Arium^®^ Pro Ultrapure Water System provided by ForteBio (Fremont, CA, USA); and pipets were purchased from Eppendorf Corporate (Hamburg, Germany).

### 4.2. Molecular Docking

#### 4.2.1. Preparing the Receptor and Ligands

The three-dimensional (3D) structure information of the mouse IgG antibody was obtained from the Protein Data Bank (PDB) whose code is 1IGT and imported into DS for removal of water molecules and ligand compounds, followed by automated dehydration, hydrogenation, loop completion, and protein optimization to achieve the most energetically favorable conformation, which served as the receptor. Simultaneously, 140 ssDNA whose 3D information has been resolved were subjected to the same treatment, primarily found in protein-DNA complexes, whereupon the protein group and redundant components in the original structure were eliminated. The optimized ssDNA structure was subsequently employed as a ligand.

#### 4.2.2. Running ZDOCK

The binding mode between mouse IgG and DNAFL was determined using the rigid body docking program ZDOCK integrated into DS; while keeping the position of the receptor protein fixed, the ligands were rotated around the receptor in a rigid-binding manner to explore potential binding sites at different grooves. The program settings included an angular step size of 15 for rotational sampling and ZRANK was set to false, in terms of clustering classification, docked poses were generated through 54,000 predictions centered around the IgG structure, which represents possible binding domains. The top 2000 poses with the highest ZDOCK scores were selected within a small cluster radius of 6.0 Å for RMSD cutoff and a smaller interface cutoff of 9.0 Å, while other parameters remained the default.

#### 4.2.3. Virtual Screening

After ZDOCK runs, the binding situations were assessed based on the ZDOCKScore, and the conformations of each cluster were filtered to determine if the ligand and antibody binding site was located within the target region (particularly in the Fc segment). The conformation with the highest score from pose1 in the molecular docking results was merged with the protein structure of mouse IgG antibody to create a virtual protein-DNAFL complex, which was considered as the optimal binding conformation. The conformation of DNAFL bound in the Fc segment was selected for further investigation.

Based on the optimal conformation of each complex, the frequency of amino acids in the IgG antibody protein sequence was determined to identify the optimal binding sites. To assess the appropriateness of the molecular docking method, mutations were introduced at the best binding sites and re-docked with DNAFLs whose ZDOCK Scores were highest, intermediate, and lowest in the 49 nucleotide sequences bound with the Fc segment.

Additionally, the contact number of amino acids and monosaccharides to the base was determined using the “Analyze Protein Interface” function in DS, enabling analysis of their contact propensity and interaction.

### 4.3. Affinity of the DNAFLs to Mouse IgG

#### 4.3.1. Affinity and Binding Sites of Virtual Screening DNAFLs to Mouse IgG

Octec K2 Molecular Interactor operates primarily based on label-free biological film interference techniques. The experimental procedures are as follows: SAX2.0 biosensors were activated using PBST-B buffer and then exposed to DNAFL solution (5′-end modified by biotin, 0.5 μM) for loading, which lasted for 10 min. Subsequently, the biosensors were sequentially immersed in different concentrations (0.0125, 0.025, 0.05, 0.1, and 0.2 mg mL^−1^) of mouse IgG solution for reaction, and each association or dissociation time was set at 5 min while PBST-B served as a negative control. Before transferring the SAX2.0 biosensors to the next concentration sample well, any antibodies bound to the surface of the biosensors during previous processes were completely dissociated using a regeneration solution. Octec K2 device analysis system enables direct determination of the association rate constant (*K_on_*) and dissociation rate constant (*K_dis_*) between DNAFLs and mouse IgG antibodies, as well as the *K_D_* value which can be directly calculated using Equation (2), and the resulting “Association-dissociation” curve is fitted. Throughout the experiment, a temperature of 30 °C was maintained with an addition of 200 uL solution into each well of a standard 96-well plate.

The binding site within the Fc segment was determined by designing a competitive binding assay between SPA and DNAFL. The membrane interference sensor ProA was activated with PBST-B buffer and incubated with mouse IgG antibody (0.2 mg mL^−1^) for 10 min, followed by binding to unmodified DNAFL (0.5 μM) for 5 min, whereafter this process of immediate dissociation for 5 min and repetition three times yielded a response signal for analysis. PBST-B was utilized as a negative control throughout the experiment while maintaining other experimental conditions constant to obtain “Association-dissociation” curves. The presence of binding sites was determined by assessing the generation of reaction signals.

#### 4.3.2. Affinity of Poly-Base to Mouse IgG

The synthetic poly-base single oligonucleotide sequence (5′-end modified with biotin0, 0.5 μM) has lengths of 10, 15, 20, 25, and 30 nt, respectively. The affinity of the poly-base to the mouse IgG antibody was assessed using identical methodologies to those described in Section 4.3.1 while maintaining consistent experimental conditions and data processing methods.

## 5. Conclusions

To address the bottleneck issue of antibody directional immobilization at the immune sensing interface, we initially selected mouse IgG antibodies whose PDB name is 1IGT as the molecular docking receptor. Subsequently, we obtained the DNAFLs from the resolved molecular structure of a protein complex containing single-stranded nucleotide chains as the ligands. The binding modes between DNAFLs and mouse IgG were simulated using rigid molecular docking technology based on the ZDOCK algorithm. By applying a scoring function, we identified 49 DNAFLs that were fully bound to the Fc segment of the antibody out of 140. Through analysis of interactions at the binding interface, key binding sites were determined including Glu423 in chains B and D, while Nag8, Man7, and Bma3 in oligosaccharide chains, and these sites were primarily located around the cavity of the Fc segment and C_H_2 domain of chain B. Then, the mutagenesis of crucial binding sites further exemplified the practicality of molecular docking. The analysis also revealed a high affinity between Glu423 and guanine along with a higher affinity for guanine and cytosine compared to thymidine and adenine among high-frequency amino acid sites within the Fc segment. Furthermore, guanine was found to be a primary target for important glycosyl sites involved in the binding surface. Additionally, through molecular affinity assay, we identified 20 DNAFLs with a strong affinity towards whole-molecule mouse IgG antibodies. Among these DNAFLs with varying lengths and base compositions (minimum *K_D_* is 2.41 × 10^−7^), those exhibiting a higher guanine/thymidine ratio demonstrated stronger affinity. Moreover, by designing poly-base nucleotide chains for IgG binding confirmation purposes, bases G and T showed a higher probability for combining antibodies while optimal performance was achieved with sequence lengths around 15–20 nt. In summary, we have successfully identified high-affinity DNAFLs that bind to the Fc segment of mouse IgG antibody, and DNAFLs with even higher affinity for specific target sites can be further designed based on these findings. These discoveries hold significant value in advancing the development of DNAFL-based immune sensing interfaces and offer a novel technical approach for designing highly sensitive immunosensors for biological detection.

## Figures and Tables

**Figure 1 ijms-25-08681-f001:**
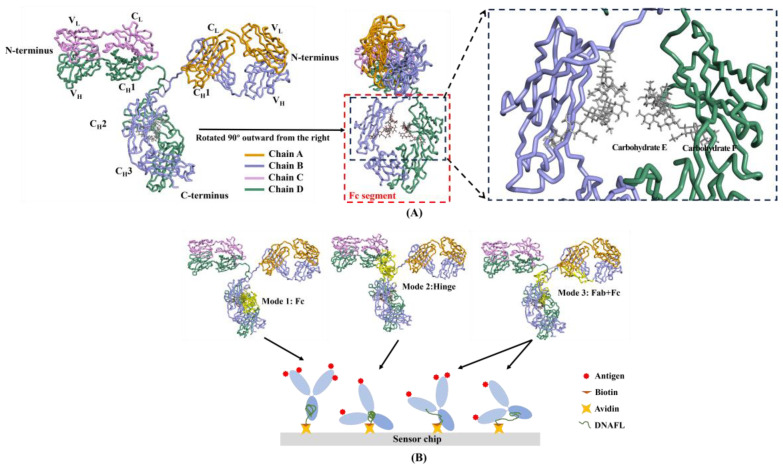
(**A**) 3D structure of mouse IgG (PDB code: 1IGT) with the light chain A and C colored in pink and orange while the heavy chain B and D is represented in blue and green (**left**), and the cavity in the middle of the Fc segment, as well as the oligosaccharide chain E and F within it can be observed by rotating the left figure 90° outward from the right (**right**); (**B**) three binding modes between DNAFL and IgG and their possible binding at the biosensing interface while DNAFLs were linked to the sensor chip using an biotin-avidin system.

**Figure 2 ijms-25-08681-f002:**
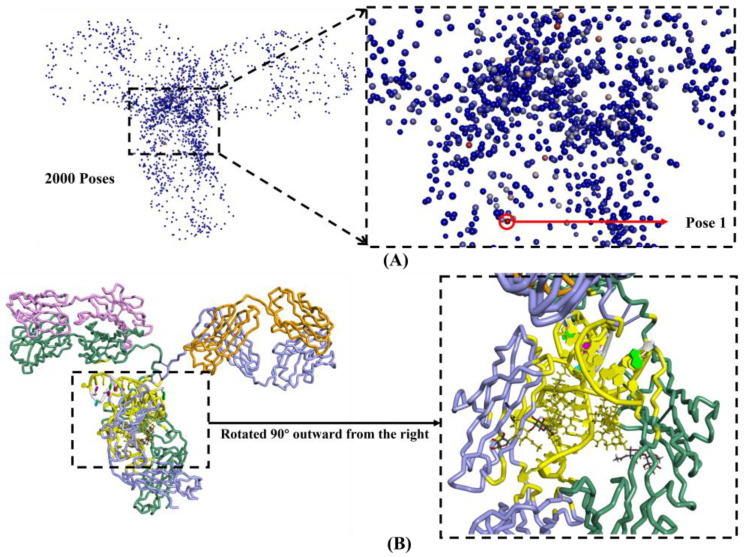
Process and outcomes of molecular docking for (**A**) distribution map of 2000 possible poses (**left**) and pose1 with the highest score (**right**) while the intensity of the red dot corresponds to a higher pose score, while the blue color indicates a lower score; (**B**) frontal view (**left**) and right-side view (**right**) of DNAFL binding sites in the Fc segment about single-stranded nucleotide from protein complex whose PDB code is 3HXQ.

**Figure 3 ijms-25-08681-f003:**
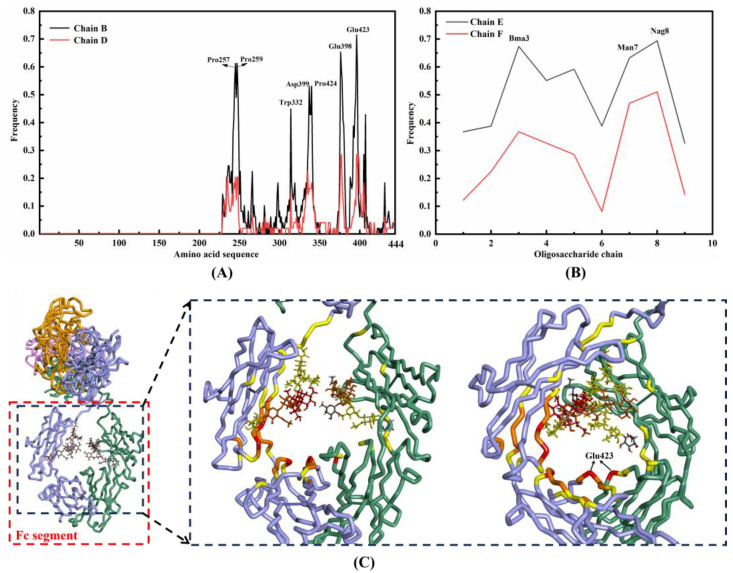
(**A**) Binding frequencies of amino acid sites in the Fc segment (the Fc segment of Chain B and D spans approximately from amino acids 230 to 444(Lys242-Arg474)); (**B**) binding frequency of monosaccharide sites in the Fc segment (Chain E and F both include 9 monosaccharides with the first eight components being identical while the 9th of E and F is furanose and fructose, respectively); (**C**) distribution of high-frequency amino acid sites in Fc segment (**left**) and position of Glu423 on chain B and D, respectively (**right**).

**Figure 4 ijms-25-08681-f004:**
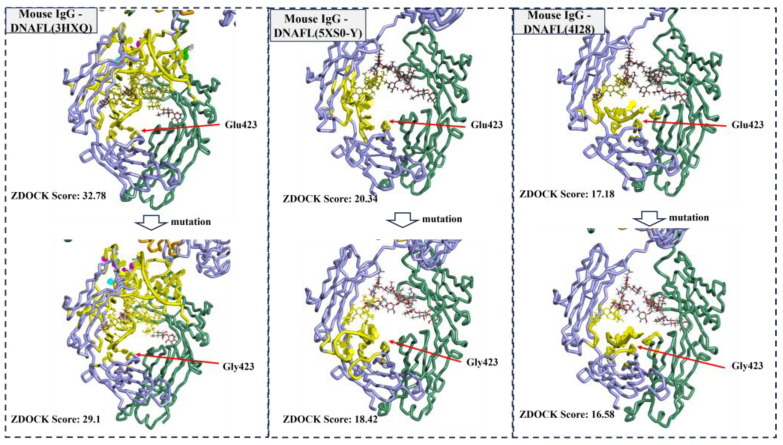
Distribution of binding sites before and after the mutation.

**Figure 5 ijms-25-08681-f005:**
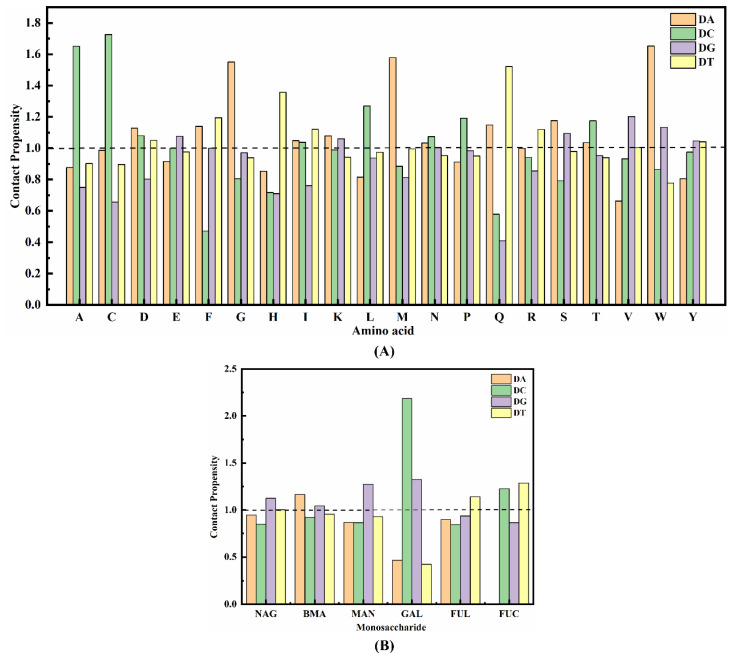
Contact propensity of (**A**) 20 amino acids and (**B**) 6 monosaccharides to bases, respectively, and the dashed line represents the scenario where the *P_ij_* = 1.

**Figure 6 ijms-25-08681-f006:**
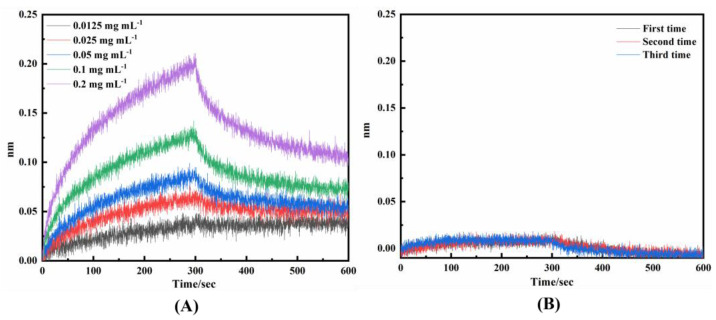
Association−dissociation curve of (**A**) DNAFL (derived from an oligonucleotide strand in a protein compound whose PDB code is 6GN7 as an example) binding to mouse IgG, and (**B**) signal response of DNAFL to SPA in competitive binding assay.

**Figure 7 ijms-25-08681-f007:**
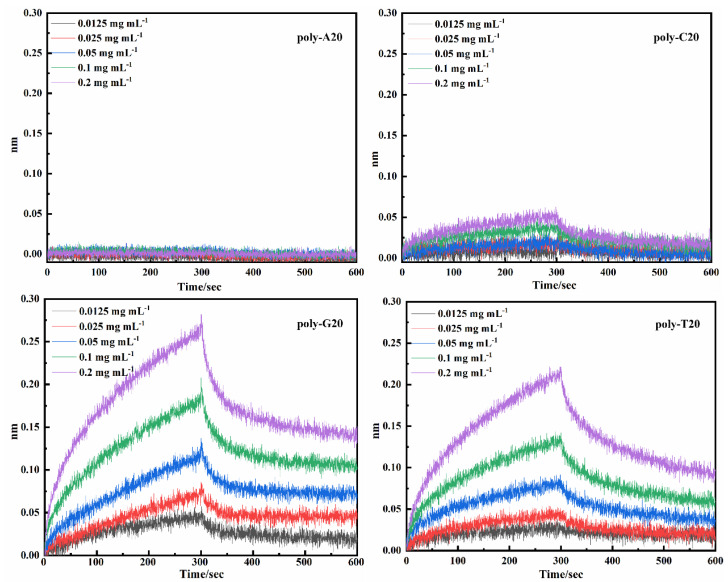
Association-dissociation curve of poly-A20, poly-C20, poly-G20 and poly-T20.

**Table 1 ijms-25-08681-t001:** *K_on_*, *K_dis_*_,_ and *K_D_* of 20 affinity DNAFLs to mouse IgG antibody and SPA-IgG.

	PDB Code	*K_on_* (1/Ms)	*K_dis_* (1/s)	*K_D_* (M)	*R* ^2^
1	5XRZ	3.76 × 10^3^	3.17 × 10^−3^	8.43 × 10^−7^	0.9402
2	6GN7	9.52 × 10^3^	2.29 × 10^−3^	2.41 × 10^−7^	0.9294
3	4GNX	1.81 × 10^3^	3.80 × 10^−3^	2.10 × 10^−6^	0.9622
4	7JSG	3.11 × 10^3^	3.48 × 10^−3^	1.12 × 10^−6^	0.9692
5	2C62	1.85 × 10^3^	6.00 × 10^−3^	3.24 × 10^−6^	0.9251
6	7XHD	6.29 × 10^3^	3.28 × 10^−3^	5.21 × 10^−7^	0.9229
7	6SKO	4.26 × 10^3^	4.54 × 10^−3^	1.07 × 10^−6^	0.9636
8	1RDE	2.91 × 10^3^	3.18 × 10^−3^	1.09 × 10^−6^	0.9661
9	2CCZ	4.42 × 10^3^	4.02 × 10^−3^	9.09 × 10^−7^	0.9614
10	6CRM	5.14 × 10^3^	5.70 × 10^−3^	1.11 × 10^−6^	0.9257
11	1PH1-D	6.97 × 10^3^	6.14 × 10^−3^	8.81 × 10^−7^	0.8856
12	1OTC	8.24 × 10^3^	3.41 × 10^−3^	4.14 × 10^−7^	0.9031
13	4JS5	3.34 × 10^3^	5.78 × 10^−3^	1.73 × 10^−6^	0.9447
14	3UGO	5.38 × 10^3^	4.27 × 10^−3^	7.93 × 10^−7^	0.8999
15	2A0I	6.39 × 10^3^	5.57 × 10^−3^	8.72 × 10^−7^	0.9134
16	2KN7	5.06 × 10^3^	6.08 × 10^−3^	1.20 × 10^−6^	0.94
17	4OU6	5.06 × 10^3^	4.79 × 10^−3^	9.47 × 10^−7^	0.9497
18	7JSI	2.58 × 10^3^	4.73 × 10^−3^	1.83 × 10^−6^	0.9672
19	5FGP	2.74 × 10^3^	4.13 × 10^−3^	1.51 × 10^−6^	0.9522
20	6S3M	6.93 × 10^3^	8.06 × 10^−3^	1.16 × 10^−6^	0.9262
21	SPA−IgG	4.80 × 10^4^	7.16 × 10^−4^	1.49 × 10^−8^	0.959

**Table 2 ijms-25-08681-t002:** *K_on_*, *K_dis,_* and *K_D_* of poly-bases DNAFLs to mouse IgG antibody.

	PDB Code	*K_on_* (1/Ms)	*K_dis_* (1/s)	*K_D_* (M)	*R* ^2^
1	poly-A10	No response			
2	poly-A15	No response			
3	poly-A20	No response			
4	poly-A25	No response			
5	poly-A30	No response			
6	poly-C10	No response			
7	poly-C15	4.27 × 10^3^	5.81 × 10^−3^	1.36 × 10^−6^	0.9117
8	poly-C20	7.42 × 10^3^	5.65 × 10^−3^	7.61 × 10^−7^	0.8997
9	poly-C25	No response			
10	poly-C30	No response			
11	poly-G10	6.87 × 10^3^	2.02 × 10^−3^	2.95 × 10^−7^	0.9301
12	poly-G15	8.42 × 10^3^	2.30 × 10^−3^	2.73 × 10^−7^	0.9676
13	poly-G20	7.85 × 10^3^	2.45 × 10^−3^	3.12 × 10^−7^	0.9583
14	poly-G25	7.24 × 10^3^	5.01 × 10^−3^	6.92 × 10^−7^	0.8785
15	poly-G30	4.39 × 10^3^	2.45 × 10^−3^	5.57 × 10^−7^	0.8844
16	poly-T10	3.79 × 10^3^	5.66 × 10^−3^	1.49 × 10^−6^	0.9438
17	poly-T15	4.42 × 10^3^	4.02 × 10^−3^	9.09 × 10^−7^	0.9614
18	poly-T20	4.50 × 10^3^	3.74 × 10^−3^	8.30 × 10^−7^	0.9628
19	poly-T25	4.18 × 10^3^	4.20 × 10^−3^	1.01 × 10^−6^	0.9661
20	poly-T30	2.50 × 10^3^	3.57 × 10^−3^	1.43 × 10^−6^	0.9448

## Data Availability

The data presented in this study are available in this article and in the Appendix A section.

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
