# Peer review of "Virtual Screening and Validation of Affinity DNA Functional Ligands for IgG Fc Segment"

_ijms, 2024, doi:10.3390/ijms25168681_

Round 1

Reviewer 1 Report

Comments and Suggestions for Authors

In this manuscript, a computational and experimental work has been implemented to find possible oligonucleotide ligands of IgG with the aim to improve the immunosensors performances. At a molecular level the authors tried to find a DNA ligand able to bind the IgG useful to immobilize antibodies on immunosensors and with steric hindrance lower than the SPA protein, which is an already known ligand of IgG.

The manuscript is well written, and the introduction is clear stated, but it should be enriched in computational and experimental methods.

For example, It seems they performed a blind docking, but if they wanted to find a DNAFL molecule able to compete with the IgG binding to SPA why not implement the X-ray structure of the complex (PDB ID 4ZNC) and use the SPA position in the complex to project a virtual space in which to screen molecules? Moreover, the study on poli bases affinity would have had more sense related to this approach, as they could have selected DNSFL molecules also on the basis of those results and having in their hands data on IgG residues binding SPA from the X-ray structure. 

This protocol should have been added to that yet performed. In addition, as they have calculated that Glu 423 is the most frequently involved in binding with DNSFL, why don’t mutate it in Gly in the IgG structure and perform the same screening? The docking score for the complexes with the mutated antibody should be worse than those whit the wild type one and this could be a way to assess the appropriateness of the docking protocol.

In addition, it is not clear what is the method used to KD calculation. Is it SPR? It should have been  specified in introduction, results and material and methods sections, and all association-dissociation curves should have been reported in the supplementary materials file. Which is the IgG-SPA KD? This value should have been used as positive control.

Line 328-329: The sentence “Compared to the other three bases, proline, being the most frequently occurring base, exhibits a higher propensity for interacting with cytosine deoxyribonucleotide” is wrong written? The proline is not a base. The same for the following sentence :” Conversely, glutamic acid, as the second most frequent base, primarily interacts with 330 guanine”.

In conclusion, in my opinion the aim of this manuscript is interesting but it must be better designed and enriched with more and better addressed computational and experimental work.

Comments on the Quality of English Language

The quality of the language is good

Author Response

Comments 1: It seems they performed a blind docking, but if they wanted to find a DNAFL molecule able to compete with the IgG binding to SPA why not implement the X-ray structure of the complex (PDB ID 4ZNC) and use the SPA position in the complex to project a virtual space in which to screen molecules? Moreover, the study on poli bases affinity would have had more sense related to this approach, as they could have selected DNSFL molecules also on the basis of those results and having in their hands data on IgG residues binding SPA from the X-ray structure.

Response 1: The objective of this study is to identify DNA functional ligands (DNAFL) that can specifically bind to the Fc segment of mouse IgG through molecular docking, and to validate the binding sites of DNAFL by investigating their competition with SPA-bound IgG. However, since only a portion of the amino acids involved in binding are engaged by SPA, it cannot fully occupy the entire Fc segment. Therefore, the binding site of DNAFL within the Fc segment may not necessarily be identical to that of SPA. Consequently, screening molecules based on virtual space projection using SPA position may not applicable in this study.

Comments 2: This protocol should have been added to that yet performed. In addition, as they have calculated that Glu 423 is the most frequently involved in binding with DNSFL, why don’t mutate it in Gly in the IgG structure and perform the same screening? The docking score for the complexes with the mutated antibody should be worse than those whit the wild type one and this could be a way to assess the appropriateness of the docking protocol.

Response 2: Thank you for sharing your perspective, and we fully concur with your suggestion. Therefore, we incorporated experiments on mutant binding sites into the Materials and Methods (lines 533-535), and subsequently included the corresponding outcomes in both the Results (lines 253-261) and Discussion (lines 425-428) to validate the rationality of the docking protocol. Additionally, we have included Figure 4 to present the docking results after mutation and compare them with the original binding position, which exhibit minimal changes and ZDOCK Scores were reduced after the docking, once again demonstrating the appropriateness of this molecular docking protocol.

Comments 3: In addition, it is not clear what is the method used to KD calculation. Is it SPR? It should have been specified in introduction, results and material and methods sections, and all association-dissociation curves should have been reported in the supplementary materials file. Which is the IgG-SPA KD? This value should have been used as positive control.

Response 3: Thank you for pointing out the incompleteness in our article. Based on your suggestion, we have incorporated the principle of molecular interaction experiment and emphasized the significance of KD in biochemistry in the introduction section (lines 138-159). Furthermore, we introduced the Eq.2 for calculating the KD (lines 298-305) in the Results, with all necessary parameters obtainable from the Octec K2 device analysis system. Affinity constant KA mentioned in the original manuscript and Figure 5(B) was omitted, considering that KD is commonly employed for affinity characterization in most of the references. Table 1 and Table 2 now presents updated parameters, including Kon, Kdis, and KD for all affinity DNAFLs, and manuscript was revised to replace all mentions of affinity with the term KD. In the Materials and Methods, we have also revised and refined the description of procedures for conducting DNAFLs affinity assay with mouse IgG (lines 555-558). Additionally, we have further updated the supplementary material to include all “Association- dissociation” curves mentioned in the manuscript, as well as the newly added curve of SPA-IgG which has been utilized as a positive control and exhibits a KD of 1.49E-08.

Comments 4: Line 328-329: The sentence “Compared to the other three bases, proline, being the most frequently occurring base, exhibits a higher propensity for interacting with cytosine deoxyribonucleotide” is wrong written? The proline is not a base. The same for the following sentence:” Conversely, glutamic acid, as the second most frequent base, primarily interacts with 330 guanine”

Response 4: Thank you for pointing out the inappropriateness “Line 328-329” and the following sentence in the original manuscript. We have revised the description of these two sentences in the manuscript where there are errors to " Compared to the other three bases, proline, being the most frequently occurring amino acid, exhibits a higher propensity for interacting with cytosine deoxyribonucleotide. Conversely, glutamic acid, as the second most frequent amino acid, primarily interacts with guanine.” which will be found in Line 438-440.

Reviewer 2 Report

Comments and Suggestions for Authors

The authors studied DNA functional ligands to Fc segments using molecular dynamics simulation, particularly with the ZDOCK algorithm. They also empirically tested the molecular affinity and the competitive binding, and found 49 (simulation) and 25 (experimental) DNAFL with greater probability to bind to IgC Fc. The results obtained by the authors are significant, and may lead to further developments on molecular sensing interfaces.

The article is well written and the results are generally well presented. The work can be accepted for publication after the authors answer some questions about the simulation adopted approximations:

1- It is stated in line 95: Since macromolecules such as protein-protein or protein-DNA/RNA complexes undergo minimal conformational changes during MD simulations [...]. This is actually an approximation, as stated in ref 28. Please, clarify this in the text or indicate a reference that corroborate this statement. Please, expando this question to the discussion of the results: how a flexible docking model could impact the obtained results?

2- The nature of the MD simulations is not clear. Please, expand the methodology to explain docking process on physical grounds, the potentialities and limits of the simulations.

The article can be accepted for publication after these questions are adequately incorporated in the text. My recommendation is that this text goes through a minor revision.

Author Response

Comments 1: Since macromolecules such as protein-protein or protein-DNA/RNA complexes undergo minimal conformational changes during MD simulations [...]. This is actually an approximation, as stated in ref 28. Please, clarify this in the text or indicate a reference that corroborate this statement. Please, expand this question to the discussion of the results: how a flexible docking model could impact the obtained results?

Response 1: Thank you for your valuable advice regarding the inaccurate description in our original article. In the original manuscript, “Since macromolecules such as protein-protein or protein-DNA/RNA complexes undergo minimal conformational changes during MD simulations [...]”, as stated in ref 28, which is intended to demonstrate that ZDOCK exhibits superior accuracy in rigid molecular docking. To provide a more precise description, we have excluded the sentence and subsequently re-elaborated on the ZDOCK docking method (lines 124-133). However, please understand that the ZDOCK algorithm utilized in this study is based on rigid docking, and therefore there is limited correlation with flexible docking. Consequently, we regret not being able to extend our discussion of the results on this aspect of the manuscript. Nevertheless, we are more than willing to address the question regarding the” how a flexible docking model could impact the obtained results?” as follows: Currently available flexible docking algorithms treat the receptor as a fixed structure while allowing for ligand binding flexibility. Nonetheless, due to the dynamic nature of protein-ligand interactions and their constant conformational changes during binding, accommodating these variations poses a challenge for flexible docking methods. Even minor conformational alterations can result in significant changes in the docking pose, leading to reduced accuracy and precision in flexible docking outcomes. However, with advancements in computational power, flexible docking has gradually emerged as an important research direction within molecular docking studies while ligands sampling and scoring functions continue to be explored and enhanced.

Comments 2: The nature of the MD simulations is not clear. Please, expand the methodology to explain docking process on physical grounds, the potentialities and limits of the simulations.

Response 2: Thanks for your valuable suggestions. We have incorporated the principles of molecular docking (MD) simulation into the introduction section, providing a detailed explanation of the MD process on physical grounds, and complementing it with potential applications and challenges (lines 88-108). We hope that these supplementary details will provide enhanced support for this research.

Round 2

Reviewer 1 Report

Comments and Suggestions for Authors

The manuscript has been improved, I have only a few suggestions for minor revisions:

The paragraphs added to the introduction section regarding molecular docking and the methods used to KD calculations are too long and better suited for a review, so, unless these addition have been suggested by another reviewer, in is better to reduce them for clarity purpose.

The achronim "MD" is not used for molecular docking but for molecular dynamics. It is better to avoid use it in the paper as it can be misunderstanding

Author Response

Comments 1: The paragraphs added to the introduction section regarding molecular docking and the methods used to KD calculations are too long and better suited for a review, so, unless these addition have been suggested by another reviewer, in is better to reduce them for clarity purpose.

Response 1: Thanks for your valuable suggestions. The methodology, docking process, potentialities and limits of molecular docking in the manuscript are the contents that need to be expanded by another reviewer. We have thoroughly considered and incorporated your feedback to simplify the description of molecular docking (lines 88-108 and lines 123-132) and KD calculation (lines137-151) in the introduction section.

Comments 2: The achronim "MD" is not used for molecular docking but for molecular dynamics. It is better to avoid use it in the paper as it can be misunderstanding.

Response 2: Thanks for sharing your perspective, we concur with your suggestion. In order to prevent any potential confusion with the expression method of molecular dynamics, we have eliminated the abbreviation MD for molecular docking in the manuscript (lines 88 and 119).
